# Development of Urdu version of Chronic Respiratory Disease Questionnaire Self-Administered Standardized (CRQ-SAS); validity and reliability analysis in COPD patients

**Momina Kashif, Danish Hassan** [ID]*, **Saira Khalid, Syed Shakil ur Rehman, Nimra Noor**

Riphah College of Rehabilitation & Allied Health Sciences, Riphah International University, Islamabad, Pakistan

* danish.hassan009@gmail.com

## Abstract

**Data Availability Statement:** Availability of data and materials: This is a translation from the original English version and is protected under copyright

### Purpose

Chronic Respiratory Disease Questionnaire Self-Administered Standardized (CRQ-SAS) is a valid and reliable tool that evaluates the health-related quality of life among the adult population affected with chronic respiratory disorders (CRDs) and has been translated into many languages as per need. The main objective of this study was to translate the CRQ-SAS into the Urdu language and evaluate its psychometric properties.

### Methodology

It was a two-staged study that consisted of translating the original version into Urdu language and then psychometric testing of the translated version. The reliability of the translated questionnaire was assessed by measuring its internal consistency, test-retest reliability, standard error of mean (SEM) & minimal detectable change (MDC). Validity was determined by evaluating its content for content validity, construct (convergent and discriminative) validity, and exploratory factor analysis. Data was analyzed using SPSS v 28 with an alpha level < 0.05 considered to be significant.

### Results

CRQ-SAS U had an excellent internal consistency (Cronbach's Alpha α = 0.89), test-retest reliability (ICC$_{2,1}$) = 0.91 of all items, and low SEM = 0.11 and MDC = 0.65. S-CVI was 0.9, with statistically significant difference across the response of COPD patients and healthy subjects, and a high degree of correlation with St Georges Respiratory Questionnaire (r = 0.7–0.9) proving CRQ-SAS U content, discriminant and convergent valid respectively. Exploratory factor analysis identified two factors responsible for 80% of the variance.

**Funding:** The author(s) received no specific funding for this work.

**Competing interests:** The authors have declared that no competing interests exist.

## Conclusion

CRQ-SAS U demonstrated optimal psychometric properties which renders it to be used in Urdu speaking populations with COPD.

## Introduction

CRDs affect 7.4% of the world's population and is attributed to the third leading cause of death besides cardiovascular problems and cancer [1]. CRDs includes a wide spectrum of pulmonary disorders but form prevalence point of view pneumonia, asthma, chronic obstructive pulmonary diseases (COPD), lung cancer, and tuberculosis are the five most common respiratory disorder worldwide [1,2]. Annually, four million people die prematurely due to CRDs [3], which make them counted among the leading causes of mortality and morbidity [4] affecting both genders across a wide spectrum of age [1]. Asian subcontinent which is 60% of the world's population was affected most with 75% of death due to CRDs from 1990–2017 posing a significant health concern [5]. Smoking (including secondhand smoke), exposure to air pollution (inside and outside), allergies, occupational exposure, poor diet, obesity and physical inactivity were among the main risk factors for CRDs [1,6]. CRDs impose a considerable financial and social cost on patients and society [7,8] and their deleterious effects result in lower engagement in professional support, poor physical and psychological health, and poor self-management [9].

Clinicians and researchers acknowledge the importance of assessing subjective health-related quality of life (HRQL) as a patient-important outcome in managing patients with CRDs [10]. There are several generally validated HRQL tools but tools that are specialized in assessing disease impact such as the Chronic Respiratory Questionnaire (CRQ) are more specific to response than generic instruments in general [11]. CRQ is the most widely used disease-specific assessment tool for assessing HRQL in chronic respiratory disease patients which was developed by Guyatt et al 1987 [12]. It was designed to be administered by an interviewer, but later on, a self-administered version was made accessible [13]. It has 20 items divided into four categories: dyspnea (5 items), fatigue (4 items), emotional function (7 items), and mastery (4 items). Patients score their experience on a 7-point Likert scale ranging from 1 (highest impairment) to 7 (no impairment) when completing this instrument [14,15]. Each domain's total score was divided by the number of items answered, providing a possible score ranging from one to seven, with larger numbers indicating greater function and vice versa [16]. The applicability of CRQ is diverse as compared to other tools like COPD assessment test (CAT) and clinical COPD questionnaire (CCQ) which are used in one condition only.

Pakistan is the fourth most populous country in the Asian subcontinent [17] with Urdu being the national language and a common means of communication for people with varied linguistic backgrounds. The prevalence of CRDs is on the rise in Pakistan [18] and it was imperative to develop an Urdu Version of this questionnaire so that the local population could benefit from it. Hence the objective of this study was to translate the Chronic Respiratory Disease Questionnaire Self-Administered Standardized (CRQ-SAS) into the Urdu language and evaluate its psychometric properties.

## Material and methods

### Sample selection

The study was conducted at Jinnah Hospital (April 22 –Oct 22), which is one of the largest tertiary care health facilities located in the metropolitan city of Lahore, Pakistan. Sample size

estimation was based on ICC value and width of confidence interval (95% CI) form previous study [19] using the formula [20] n = 16p(1−p)/w$^2$, where "p" was lowest expected ICC (0.70) and "w" was the maximum width of 95%CI (0.178). The minimum sample size estimated was 87, but we recruited a larger sample of 124 subjects (62 COPD & 62 healthy) to account for expected dropout at retest interval. COPD patients visiting the outpatient department of pulmonology of either gender, diagnosed with COPD as per GOLD criteria [21], able to speak and read Urdu language and can complete the Urdu version of CRQ-SAS within one session on a self-report basis were included in this study [16,22]. Age matched healthy individual recruited were either the attendants/family members accompanying patients in Jinnah Hospital or were employees of Riphah International University. The study was approved by the Research & Ethics Committee of Riphah International University (REC/RER & AHS/22/0317) and additional permission was also sought form Jinnah Hospital, Lahore, Pakistan. Aim of the study was explained to each participant in detail before the distribution of the questionnaire. An informed written consent form was also signed by each participant.

## Translation and validation process

Before the beginning of the study, the original author's permission (McMaster University, Hamilton, Ontario, Canada) was sought for this translation via email. The study was carried out in two stages; stage I involved the translation of the original English version of CRQ-SAS into Urdu language as per Beaton guidelines [23] and stage II involved the psychometric evaluation of the Urdu version of CRQ-SAS.

**STAGE I: Development of the Urdu version of CRQ-SAS (CRQ- SAS U).** In order to achieve linguistic equivalence between the original version and the translated version, the CRQ-SAS U was developed using a five-step (A-E) process in accordance with the recommendations made by McMaster University.

## A. Forward translation into the target language

After the conceptual analysis of the source questionnaire provided by McMaster University, CRQ-SAS was first translated from English to the Urdu language by two experts fluent in both Urdu and English languages. Each of them created a unique profile of written reports identifying the language challenges encountered during translation and justifying the use of words and phrases. Afterward, two Urdu versions of the CRQ-SAS were reconciled to create a single Urdu questionnaire while still considering the original form.

## B. Backward translation into the original language

Two independent, native English-speaking bilingual translators back-translated the reconciled Urdu version of CRQ-SAS into the English language independently. They had no prior medical background and were totally unaware of the questionnaire's concept. The two back-translated versions were reconciled to develop a single English version of CRQ-SAS.

## C. Creator Review of Back Translated version of CRQ-SAS

The back-translated version of CRQ SAS was reviewed by the creator which worked to improve the semantic, idiomatic, experiential, and conceptual equivalence of all prior versions of the Urdu and English translations with the help of translators.

## D. Cognitive Debriefing

Cognitive debriefing was performed by administering this questionnaire to five patients recruited from the same clinical settings with COPD. Patients were asked to highlight any difficult or incomprehensible word or any sentence making the question complicated. Solutions were given so that the questions may be understood better. The cognitive debriefing data was then reviewed and approved by the creator of the original questionnaire.

## E. Proofreading and finalization of the translation

Final CRQ-SAS U questionnaire was created with the help of the expert committee (author, two cardiopulmonary physical therapists & one pulmonologist) which worked to improve the semantic, idiomatic, experiential, and conceptual equivalence of all prior versions of the Urdu and English translations. After rigorous proofreading of CRQ-SAS U along with all related documents (including translational certificates) were submitted to McMaster University.

**STAGE II: Psychometric testing of the CRQ-SAS U.** After re-consent from McMaster University, the CRQ-SAS U was prepared for psychometric testing, which involved testing it for validity and reliability. Two cardiopulmonary physical therapists with at least 2 years' experience in indoor clinical settings administered CRQ-SAS U. Both physical therapists underwent a formal training of 2 hours in which they are guided of all the process of interview and documentation of responses of COPD patients on CRQ-SAS U.

## Reliability

**Internal Consistency;** Cronbach's alpha was used to assess the internal consistency of the CRQ-SAS U. Value of Cronbach's alpha >0.8 is considered as good [24]. Item total correlation was also calculated to determine the homogeneity between each item using Pearson correlation coefficient.

**Test-Retest Reliability;** CRQ-SAS U was administered to the same patients after time interval of 7 days to determine the stability of responses. ($ICC_{2,1}$) using two-way mixed analysis of variance model [25] was measured for $1^{st}$ and $2^{nd}$ administration of CRQ-SAS U. ICC values above 0.8 or 0.9 show good or excellent reliability and ICC values below 0.5 show poor reliability [26].

**SEM & MDC;** Agreement between the $1^{st}$ and $2^{nd}$ values of CRQ-SAS U were also estimated through SEM and MDC.

The formulas used to calculate SEM and MDC are $SEM = SD \times \sqrt{1 - ICC}$ and $MDC = 1.96 \times \sqrt{2} \times SEM$, respectively. Smaller values of SEM indicated a greater degree of reliability. MDC less than 30% was considered to be acceptable [27].

**Bland and Altman (B&A) Plot:** The level of within-subject variance and limits of agreement with 95% confidence intervals were assessed using the B&A plot. The B&A plot uses the limits of agreement from the mean difference to graphically represent and quantify the agreement between two quantitative measurements. Only the agreement intervals are defined in the plot; their acceptability is not addressed. Using a basic calculation based on the mean and standard deviation of two measurements, B&A developed agreement limitations. These statistical limits are computed from the mean and standard deviation of two measurements. In the B&A plot, a graph was used to determine whether differences and other things were normal [28].

**Floor and Ceiling Effect:** The floor and ceiling effect was also calculated during the analysis by estimating the proportion of subjects that scored highest and lowest respectively [29]. If more than 15% of the patients received the highest or lowest possible score out of a potential total score, then these effects are considered to be present [28].

## Validity

**Face Validity;** A group of 10 COPD patients and 5 clinical experts panel (3 pulmonologists and 2 physical therapists) were invited to evaluate the face validity of the CRQ-SAS U. The finalized version was administered to patients and clinical expert panel and were interviewed one by one to determine the relevance and appropriateness of each item.

**Content Validity;** Finalized version of the CRQ-SAS U questionnaire was administered to the clinical expert's panel (as described above) and they were asked to rate each item in the questionnaire on a 4-point Likert scale for relevance, simplicity, clarity, and ambiguity. Item content validity index (I-CVI) was calculated by determining experts in agreement divided by the total number of experts. Scale content validity index (S-CVI) was then calculated by averaging I-CVI of all items in CRQ-SAS U. CVI > 0.80 is consider acceptable as per guideline in previous studies [24].

**Convergent validity;** Convergent validity was determined by correlating the score of the CRQ-SAS U with the St Georges Respiratory Questionnaire (SGRQ). The Pearson correlation coefficient was calculated to assess whether there was a significant correlation between these two questionnaires. Value of r between ±0.7–1.0 were interpreted as indicating strong association [30].

**Discriminative validity;** Independent sample t-test was applied to each item of the CRQ-SAS U COPD patients and healthy subjects to determine the difference in score across healthy and diseased population. P value$<$0.05 indicates a significant difference.

**Factor analysis.** Kaiser-Meyer-Olkin (KMO) and Bartlett's tests were used to evaluate the efficacy of the data. KMO test investigates sampling adequacy. The KMO value varies from 0 to 1. The desired value must be $> 0.5$ for factor analysis to be carried out [31]. Another measure of sample adequacy is the Bartlett's test of sphericity, which assesses the overall significance of all correlations among all measuring instrument items. P-value must be $< 0.05$ for proving the test's significance [32].

## Data analysis

The data was entered and analyzed using IBM SPSS v 28 (IBM Corp., Armonk, NY, USA) statistical software. p-value $< 0.05$ was considered significant. In the descriptive analysis, mean and standard deviation was reported for continuous variables while frequency and percentage for the discrete variable.

## Results

### Psychometric analysis

The details of patients and aged-matched healthy subject's demographic and clinical characteristics are listed in **Table 1**. During the process of cognitive debriefing no item was reported as difficult to interpret by the patients. COPD patients of all four grades were included.

### Reliability analysis

The mean score of items of CRQ SAS U ranged from 2.26–7.32. There was no missing response to any item. No floor and ceiling effect was noted in the responses of CRQ-SAS U. The maximum floor effect and ceiling effect was noted in Item I (11.29%) and Item 2(9.67%) respectively, which were under the acceptable limit of 15% (**Table 2**). Cronbach's Alpha for all 20 items of CRQ SAS U was 0.89 ($\alpha$ = 0.89) indicating excellent internal consistency of the translated questionnaire (**Table 3**). SEM and SDC of all items ranged between 0.13–0.35 and 0.29–0.78 respectively. However, SEM and SDC for CRQ-SAS Urdu version total score were

**Table 1. Demographic and health profile of COPD and healthy subjects.**

| Variables | | COPD (n = 62) | Healthy (n = 62) | P value |
|---|---|---|---|---|
| Age (years) | | 59.03 ± 7.79 | 54.32 ± 5.11 | <0.05[a] |
| Gender %(n) M/F | | 35.5 (22) / 64.5 (40) | 50 (31) / 50 (31) | <0.05[b] |
| Height (m) | | 1.61 ± 0.95 | 1.68 ± 1.14 | <0.05 [a] |
| Weight (kg) | | 70.8 ± 10.01 | 69.16 ± 10.60 | 0.32 [a] |
| BMI (kg/m$^2$) | | 27.11 ± 3.06 | 24.31 ± 2.76 | <0.05 [a] |
| Grade of COPD %(n) | Grade I | 4.8 (3) | ---------- | NA |
| | Grade II | 45.2 (28) | ---------- | NA |
| | Grade III | 37.1 (23) | ---------- | NA |
| | Grade IV | 12.9 (8) | ---------- | NA |
| FEV1% | | 50.59 ± 15.90 | ---------- | NA |
| FEV1:FVC Ratio | | 79.56 ± 12.29 | ---------- | NA |
| MRC Dyspnea Scale | | 2.92 ± 1.15 | ---------- | NA |

[FEV1: Forced Expiratory Volume 1$^{st}$ Second][FVC: Forced Vital Capacity][MRC: Medical Research Council][a: Independent Sample T Test][b: Chi-square Test]

0.11 and 0.65 respectively. Item total correlation of all items ranged from 0.8–0.9 except for Item 3,4 & 5 which was less than 0.5 indicating good homogeneity between the items of CRQ-SAS U (**Table 4**). Excellent test retest reliability was observed between the 1$^{st}$ and 2$^{nd}$ response among 62 COPD patients with ($ICC_{2,1}$) of individual items ranges form ($ICC_{2,1}$) = 0.81–1.00 with an average of ($ICC_{2,1}$) = 0.91 of all items of CRQ-SAS U (**Table 3**). Minimal intra-subject differences were recorded on the B&A plot between the two administration of CRQ SAS U supporting the ICC values obtained (**Fig 1**).

**Table 2. Descriptive data, distribution of responses and floor and ceiling effect (n = 62).**

| CRQ-SAS U | Mean | SD | Highest Score | Lowest Score | Missing Response to an Item | Floor (%) | Ceiling (%) |
|---|---|---|---|---|---|---|---|
| Q 1 | 2.56 | 1.25 | 6.00 | 1.00 | 0 | 11.29 | 6.41 |
| Q 2 | 2.40 | 1.57 | 7.00 | 1.00 | 0 | 4.82 | 9.67 |
| Q 3 | 6.05 | 2.54 | 8.00 | 1.00 | 0 | 0 | 1.61 |
| Q 4 | 5.66 | 2.74 | 8.00 | 1.00 | 0 | 1.61 | 4.83 |
| Q 5 | 7.32 | 1.64 | 8.00 | 1.00 | 0 | 4.82 | 0 |
| Q 6 | 2.73 | 1.28 | 6.00 | 1.00 | 0 | 0 | 0 |
| Q 7 | 2.40 | 1.40 | 7.00 | 1.00 | 0 | 0 | 1.61 |
| Q 8 | 2.53 | 1.29 | 6.00 | 1.00 | 0 | 0 | 0 |
| Q 9 | 2.95 | 1.37 | 7.00 | 1.00 | 0 | 0 | 1.61 |
| Q 10 | 2.50 | 1.13 | 5.00 | 1.00 | 0 | 0 | 0 |
| Q 11 | 2.42 | 1.09 | 5.00 | 1.00 | 0 | 0 | 0 |
| Q 12 | 2.42 | 1.18 | 6.00 | 1.00 | 0 | 0 | 0 |
| Q 13 | 2.39 | 1.19 | 6.00 | 1.00 | 0 | 0 | 0 |
| Q 14 | 2.32 | 1.11 | 6.00 | 1.00 | 0 | 0 | 0 |
| Q 15 | 2.68 | 1.40 | 6.00 | 1.00 | 0 | 0 | 0 |
| Q 16 | 2.58 | 1.43 | 6.00 | 1.00 | 0 | 0 | 0 |
| Q 17 | 2.50 | 1.34 | 6.00 | 1.00 | 0 | 0 | 0 |
| Q 18 | 2.26 | 0.97 | 5.00 | 1.00 | 0 | 0 | 0 |
| Q 19 | 2.44 | 1.30 | 6.00 | 1.00 | 0 | 0 | 0 |
| Q 20 | 2.52 | 1.46 | 6.00 | 1.00 | 0 | 0 | 0 |

**Table 3. Agreement of repeated measurements, test-retest reliability, internal consistency and item total correlation values for CRQ-SAS U (n = 62 Patients).**

| CRQ-SAS U | First Administration (Mean ± SD) | Follow up Administration (Mean ± SD) | SEM | MDC | ICC (95% CI) | Cronbach Alpha | Item Total Correlation |
|---|---|---|---|---|---|---|---|
| Q 1 | 2.56 ± 1.25 | 2.56 ± 1.25 | 0.16 | 0.35 | 1.000 (1.000,1.000) | NA | 0.829 |
| Q 2 | 2.40 ± 1.57 | 2.40 ± 1.57 | 0.20 | 0.45 | 1.000 (1.000,1.000) | NA | 0.876 |
| Q 3 | 6.05 ± 2.54 | 6.05 ± 2.54 | 0.32 | 0.72 | 1.000 (1.000,1.000) | NA | -0.328 |
| Q 4 | 5.66 ± 2.74 | 5.66 ± 2.74 | 0.35 | 0.78 | 1.000 (1.000,1.000) | NA | -0.348 |
| Q 5 | 7.32 ± 1.64 | 7.32 ± 1.64 | 0.21 | 0.47 | 1.000 (1.000,1.000) | NA | -0.442 |
| Q 6 | 2.73 ± 1.28 | 3.02 ± 1.32 | 0.17 | 0.37 | 0.942 (0.904,0.956) | NA | 0.873 |
| Q 7 | 2.40 ± 1.40 | 2.95 ± 1.36 | 0.18 | 0.39 | 0.892 (0.820,0.935) | NA | 0.802 |
| Q 8 | 2.53 ± 1.29 | 2.84 ± 1.44 | 0.17 | 0.39 | 0.921 (0.870,0.953) | NA | 0.909 |
| Q 9 | 2.95 ± 1.37 | 3.06 ± 1.33 | 0.17 | 0.38 | 0.867 (0.780,0.920) | NA | 0.823 |
| Q 10 | 2.50 ± 1.13 | 2.97 ± 1.17 | 0.15 | 0.33 | 0.845 (0.742,0.906) | NA | 0.858 |
| Q 11 | 2.42 ± 1.09 | 2.74 ± 1.20 | 0.15 | 0.33 | 0.914 (0.857,0.948) | NA | 0.802 |
| Q 12 | 2.42 ± 1.18 | 2.98 ± 1.44 | 0.17 | 0.37 | 0.892 (0.821,0.935) | NA | 0.854 |
| Q 13 | 2.39 ± 1.19 | 2.92 ± 1.32 | 0.16 | 0.36 | 0.818 (0.699,0.891) | NA | 0.895 |
| Q 14 | 2.32 ± 1.11 | 2.89 ± 1.31 | 0.15 | 0.34 | 0.898 (0.831,0.939) | NA | 0.879 |
| Q 15 | 2.68 ± 1.40 | 3.00 ± 137 | 0.18 | 0.39 | 0.848 (0.748,0.909) | NA | 0.806 |
| Q 16 | 2.58 ± 1.43 | 3.21 ± 1.46 | 0.18 | 0.41 | 0.813 (0.690,0.887) | NA | 0.883 |
| Q 17 | 2.50 ± 1.34 | 2.89 ± 1.40 | 0.17 | 0.39 | 0.870 (0.785,0.922) | NA | 0.847 |
| Q 18 | 2.26 ± 0.97 | 2.56 ± 1.05 | 0.13 | 0.29 | 0.836 (0.729,0.901) | NA | 0.865 |
| Q 19 | 2.44 ± 1.30 | 2.77 ± 1.37 | 0.17 | 0.38 | 0.871 (0.786,0.922) | NA | 0.861 |
| Q 20 | 2.52 ± 1.46 | 2.76 ± 1.29 | 0.17 | 0.39 | 0.817 (0.697,0.890) | NA | 0.874 |
| ALL Items | 3.08 ± 0.86 | 3.37 ± 0.90 | 0.11 | 0.65 | 0.949 (0.916,0.969) | 0.896 | NA |

**Table 4. Factor loading values.**

| CRQ-SAS U | Factor 1 | Factor 2 |
|---|---|---|
| Q.1 | .850 | xxxxxxx |
| Q.2 | .910 | xxxxxxx |
| Q.3 | xxxxxxx | .896 |
| Q.4 | xxxxxxx | .837 |
| Q.5 | xxxxxxx | .436 |
| Q.6 | .917 | xxxxxxx |
| Q.7 | .840 | xxxxxxx |
| Q.8 | .974 | xxxxxxx |
| Q.9 | .937 | xxxxxxx |
| Q.10 | .906 | xxxxxxx |
| Q.11 | .780 | xxxxxxx |
| Q.12 | .885 | xxxxxxx |
| Q.13 | .987 | xxxxxxx |
| Q.14 | .915 | xxxxxxx |
| Q.15 | .837 | xxxxxxx |
| Q.16 | .946 | xxxxxxx |
| Q.17 | .869 | xxxxxxx |
| Q.18 | .881 | xxxxxxx |
| Q.19 | .899 | xxxxxxx |
| Q.20 | .922 | xxxxxxx |

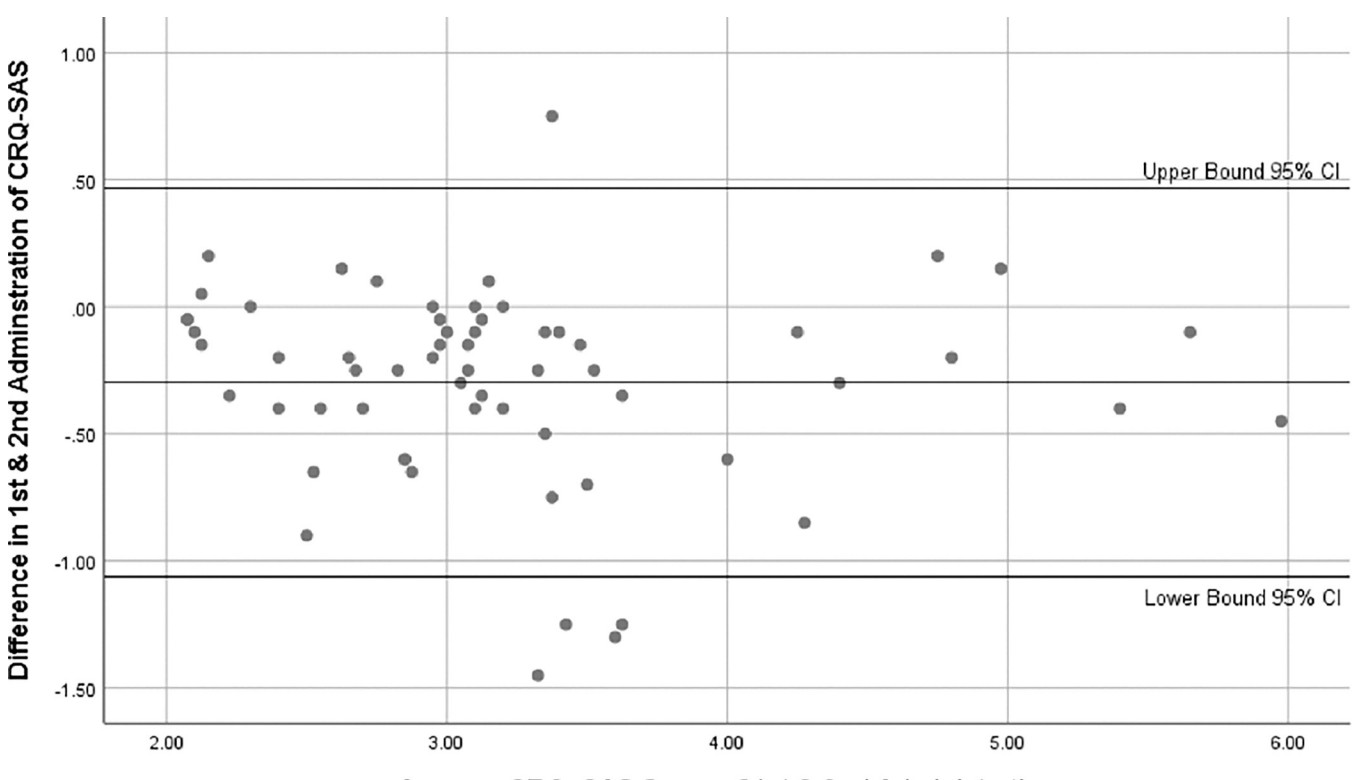

**Fig 1. Bland-Altman plot for assessing the limits of agreement and with-in subject variation.**

### Factor analysis

The factor structure of the CRQ-SAS Urdu version was evaluated through factor analysis. Bartlett's test of sphericity was found to be significant (P < 0.05) and KMO value of 0.958 showed that the sample recruited in this study was adequate. Two factors structure of the CRQ-SAS Urdu version was established based on eigenvalues > 1 (**Table 4, Figs 2 and 3**) explaining 80% of the variance.

### Validity analysis

Content validity Index of individual item (I-CVI) ranged from I-CVI = 0.8–1.0, with S-CVI = 0.9 proving CRQ-SAS U relevant to the targeted construct. Significant differences across the response of COPD patients and healthy subjects to CRQ-SAS U in all subdomains indicated discriminative validity (**Table 5**). All domains of CRQ-SAS U and St George Respiratory Questionnaire had Pearson's correlation coefficient 'r' ranging from 0.7–0.9 with a p value < 0.05 indicating significant correlation (**Table 5**).

### Discussion

The main objective of this study was to translate the CRQ-SAS into the Urdu language and evaluate its psychometric properties. As per our knowledge, this is the first research of its kind that developed the Urdu version of this questionnaire and also proved it to be valid and reliable to use. CRQ-SAS is regarded as a valid and reliable instrument since it measures health-related quality of life across several languages and adult groups with CRDs. A total of 124 individuals

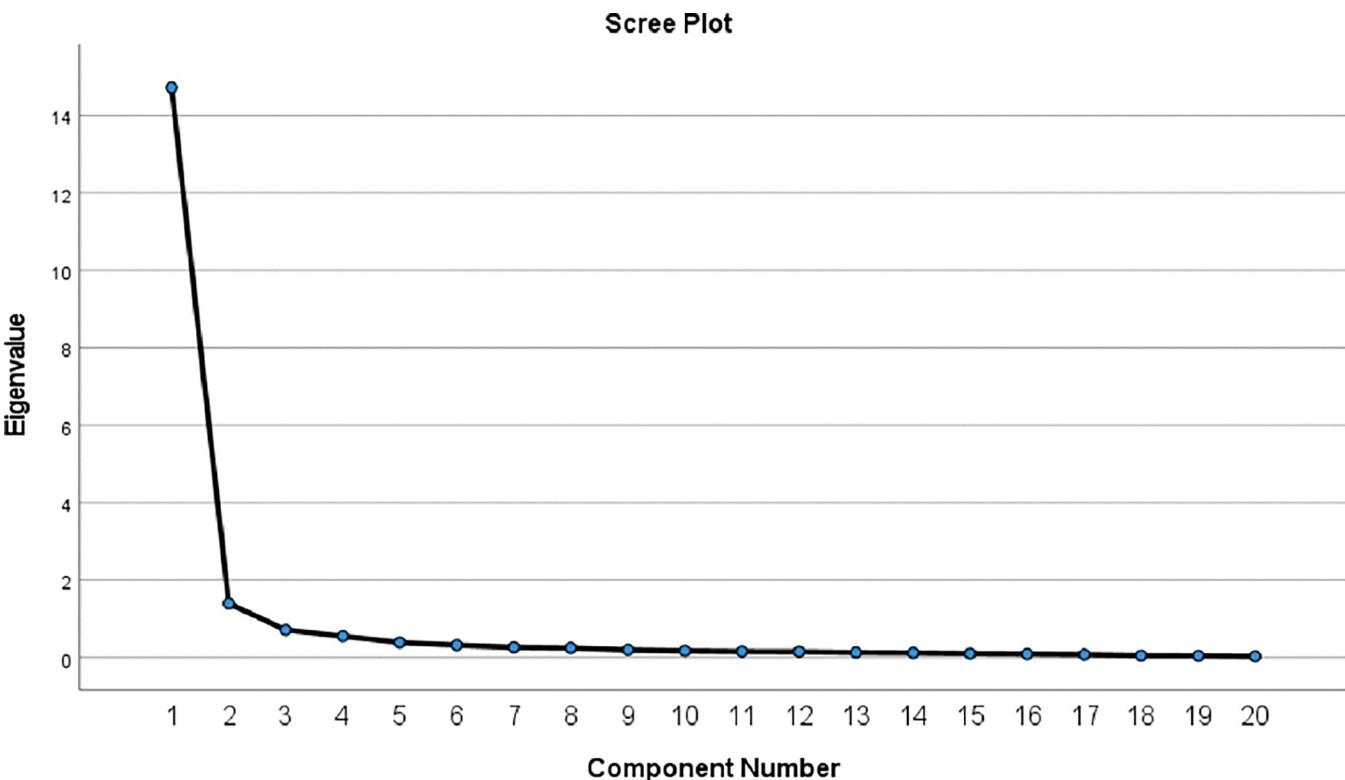

**Fig 2. Screen plot eigenvalue of components greater than 1.**

were included in the study according to inclusion and exclusion criteria, out of which 62 were COPD patients and the other 62 were healthy individuals. This study was conducted on adult COPD patients and the number of diseased patients enrolled was higher than the previous studies [33–36] except for a Spanish study [37] that tested CRQ SAS in a sample of 280 chronic respiratory diseased adolescents.

Current study found excellent test-retest reliability (stability over time) of the CRQ-SAS U. The findings of current study (ICC = 0.94) varied to those of other language translations of CRQ-SAS such as Arabic (ICC = 0.97), Malay (ICC >0.80), German (ICC >0.70), and the original language (ICC = 0.83–0.95) [16,38–40]. These differences can be attributed to variation in the test-retest interval that ranged from 2 to 5 weeks in previous studies compared to 1 week in the current study. CRQ-SAS U has a good internal consistency with a Cronbach's alpha (α) of 0.89 which was higher compared to Taiwanese [34], Malay [35], and German [19] versions that reported α = 0.80. Item total correlation values ranged between -0.328–0.909 which is also confirming the internal consistency of the CRQ-SAS Urdu version, similar to the German version with item-total correlation values ranging between 0.37–0.85 [40]. CRQ SAS contains questions that inquires information related to the impact of disease on activities. Among these questions, item 3 (walking), item 4 (performing chores such as housework, shopping, groceries) & item 5 (participating in social activities) are directly linked to participation [41]. As the socioeconomic status and daily roles of the participants was not considered, this might be the possible explanation for low item to total correlation of these three items only. No previous study had reported SDC or SEM as a measure of reliability or consistency.

SGRQ was used to determine the convergent validity of CRQ-SAS U that showed significantly high correlation in all subdomains consistent with the finding of Malay Version [35].

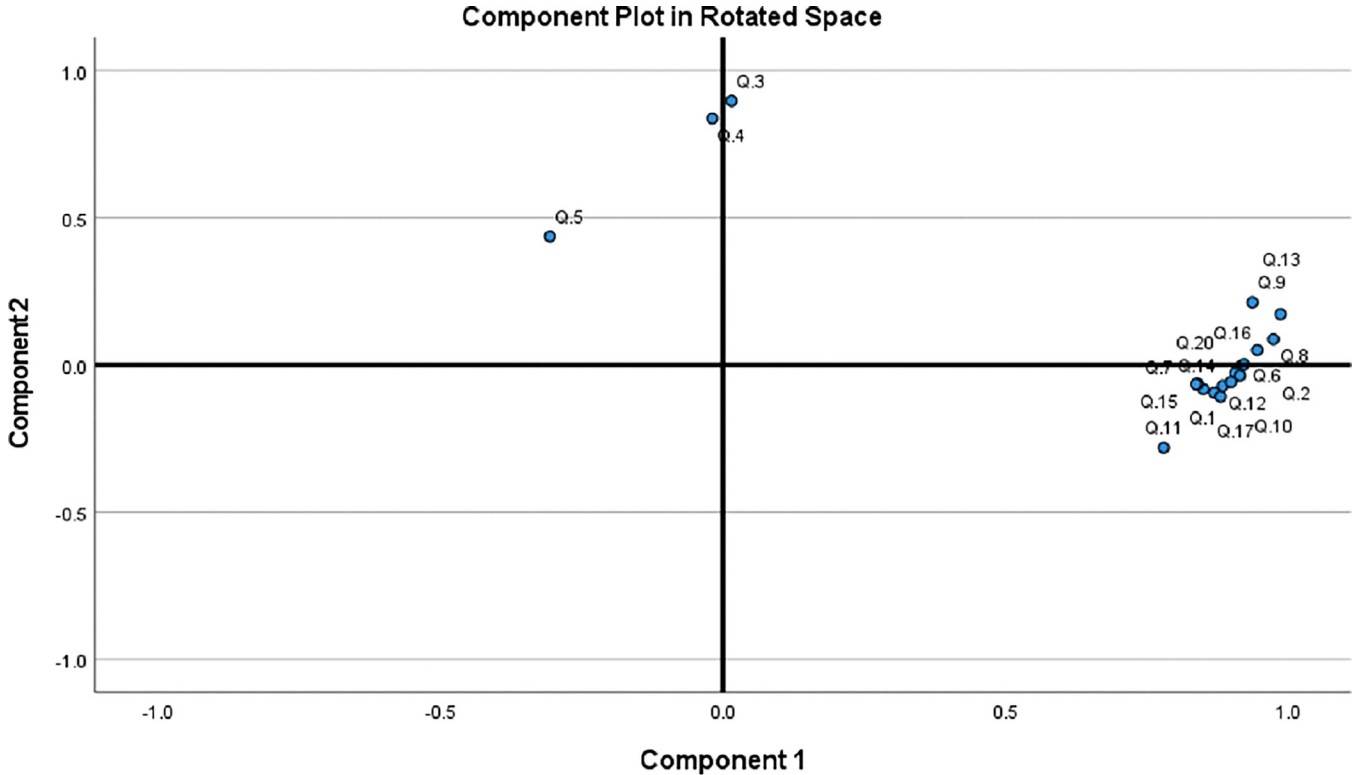

**Fig 3. Loading plots of two factors.**

**Table 5. Testing CRQ-SAS U construct validity and discriminative validity.**

| CRQ-SAS U Domains | COPD (n = 62) Mean ± SD | Healthy (n = 62) Mean ± SD | Mean Difference (95% CI) |
|---|---|---|---|
| **Dyspnea Domain** | 4.80 ± 0.95 | 6.89 ± 0.65 | 2.09* (1.80,2.38) |
| **Fatigue** | 2.53 ± 1.19 | 5.81 ± 0.99 | 3.28* (2.89, 3.67) |
| **Emotional Function** | 2.53 ± 1.12 | 5.80 ± 0.91 | 3.27* (2.91, 3.64) |
| **Mastery** | 2.43 ± 1.15 | 6.04 ± 0.77 | 3.61* (3.26, 3.96) |
| **Pearson's Correlation Coefficient Between** | | | **r** |
| **CRQ-SAS U—Dyspnea Domain** | **St Georges–Symptoms** | | 0.749* |
| | **St Georges–Activity** | | -0.751* |
| | **St Georges–Impact** | | -0.717* |
| **CRQ-SAS U—Fatigue** | **St Georges–Symptoms** | | 0.920* |
| | **St Georges–Activity** | | -0.899* |
| | **St Georges–Impact** | | -0.927* |
| **CRQ-SAS U—Emotional Function** | **St Georges–Symptoms** | | 0.935* |
| | **St Georges–Activity** | | -0.902* |
| | **St Georges–Impact** | | -0.936* |
| **CRQ-SAS U—Mastery** | **St Georges–Symptoms** | | 0.951* |
| | **St Georges–Activity** | | -0.921* |
| | **St Georges–Impact** | | -0.951* |

[* p value < 0.001].

6-minute walk distance (6MWD), short form-36 (SF-36), graded exercise test, and pulmonary function test were the other outcomes used to prove convergent validity in previous studies [33,34]. SGRQ is generic self-reported measure like CRQ SAS that can be easily used to determine the impact of CRDs on HRQol. Unlike other subjective measures e.g. CAT or CCQ [42], this questionnaire is freely available and requires no permission/license to use. SGRQ also has similar subdomains like CRQ SAS and hence was also preferred over SF-36 which is a nonspecific QOL measure. Since the data was collected form the outpatient department with very small-time margin of patient interaction, the use of objective measures like 6MWD, graded exercise test, and pulmonary function test was not feasible.

Discriminant validity was only reported in the current study that showed significant difference in the response of CRQ-SAS U in COPD patients and aged-matched healthy subjects. KMO measure of sampling adequacy was adequately high (0.958) and Bartlett's test of sphericity was found to be significant ($p < 0.05$). In a previous study, KMO and Bartlett's tests were used to evaluate the efficacy of the data and both yielded satisfactory results [43]. In a different study, the instrument was subjected to factor analysis, and the results led to the conclusion that the generated matrix could be factorized because the KMO index, which was first determined, came out to be 0.92 [44]. In the current study, two factors were responsible for the variance in data with a eigenvalue greater than 1 similar to a previous research [45]. All these results indicate that CRQ-SAS U is a very reliable and convenient tool for assessing the health status of COPD patients and can be used effectively by native pulmonologist, physicians, cardiopulmonary physical therapist and nurses to accurate measure the progression of disease and modify their management accordingly.

## Limitations

Since CRDs includes a wide range of disorders affecting pulmonary conditions, psychometric testing of CRQ-SAS U was only done on COPD patients based on its high prevalence in Pakistan. The use of this questionnaire in other chronic respiratory disorders like asthma, tuberculosis, and COVID 19 remains to be ascertained by future researchers. Responsiveness of the CRQ-SAS U was also not measured as the disease related quality of life in COPD patients was not assessed in relation to any specific treatment. Further research also need to employ Rasch analysis, which takes into account the various individual characteristics of respondents, to illustrate the connection between the responses to each item.

## Conclusion

The CRQ-SAS U questionnaire is a valid and reliable tool be used in Urdu speaking population. As it is simple and easy to understand and comprehend in Urdu language hence, we recommend health care professionals to use this tool to measure the disease-related quality of life in native COPD patients.

## Supporting information

**S1 File.**
(ZIP)

## Author Contributions

**Conceptualization:** Momina Kashif.

**Data curation:** Momina Kashif, Nimra Noor.

**Methodology:** Saira Khalid, Syed Shakil ur Rehman.

**Project administration:** Saira Khalid.

**Supervision:** Danish Hassan.

**Validation:** Nimra Noor.

**Writing – original draft:** Danish Hassan, Syed Shakil ur Rehman, Nimra Noor.

**Writing – review & editing:** Danish Hassan.

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
