## [Decision Letter · Decision Letter 0]

19 Jul 2023

PONE-D-23-14764Development of Urdu Version of Chronic Respiratory Disease Questionnaire Self-Administered Standardized (CRQ-SAS); Validity and Reliability AnalysisPLOS ONE

Dear Dr. Hassan,

Thank you for submitting your manuscript to PLOS ONE. After careful consideration, we feel that it has merit but does not fully meet PLOS ONE’s publication criteria as it currently stands. Therefore, we invite you to submit a revised version of the manuscript that addresses the points raised during the review process.

We look forward to receiving your revised manuscript.

Kind regards,

Gurkan Gunaydin

Academic Editor

PLOS ONE

5. Please upload a copy of Figure 3, to which you refer in your text on page 10. If the figure is no longer to be included as part of the submission please remove all reference to it within the text.

Additional Editor Comments:

Dear Authors,

I am pleased to inform you that three reciewers opinions have been completed for your article. All reviewers have expressed the view that your article will make a valuable contribution to the literature. However, they have also provided some suggestions for improving the overall quality of your work. Therefore, I kindly request you to compose a response letter addressing the comments of all the reviewers. Once the necessary corrections have been made, I recommend obtaining English editing for the article to ensure its general readability and language accuracy.

Sincerely

Reviewers' comments:

Reviewer's Responses to Questions

**Comments to the Author**

1. Is the manuscript technically sound, and do the data support the conclusions?

Reviewer #1: Partly

Reviewer #2: Yes

Reviewer #3: Yes

2. Has the statistical analysis been performed appropriately and rigorously? 

Reviewer #1: Yes

Reviewer #2: Yes

Reviewer #3: Yes

3. Have the authors made all data underlying the findings in their manuscript fully available?

Reviewer #1: Yes

Reviewer #2: Yes

Reviewer #3: Yes

4. Is the manuscript presented in an intelligible fashion and written in standard English?

Reviewer #1: Yes

Reviewer #2: Yes

Reviewer #3: Yes

5. Review Comments to the Author

Reviewer #1: Dear Authors,

Thanks for the effort you put into your work. I consider that the study in question will be appropriate for publication after a few minor revisions. I recommend that you reconsider on a language basis, especially for some word usages. With this; The suggestions that I can make for you valuable researchers are listed below.

It can be given in the title as translation and its validity and reliability in COPD patients. For example; Translation of Chronic Respiratory Disease Questionnaire Self-Administered Standardized (CRQ-SAS) to Urdu Language and Validity and Reliability in COPD.

Abstract

Write the long versions of the abbreviations for the first use. Such as COPD. In addition, if you have chosen an abbreviation, you should prefer the abbreviation rather than the clear version throughout the text, like CRD in discussion part of article.

Prefer to use CRQ-SAS U everywhere. In some places it remained CRQ SAS U. Indicate at first use that U is the Urdu version.

Minimal Detectable Change may be a more accurate use instead of Smallest Detectable Change.

I suggest that you review the overall text in terms of language and spelling. You have used the 2nd – 3rd meanings of some words.

Introduction

I suggest you replace some of the words you use with more common uses. For instance; “involve, contain” instead of “encompass”; “disorders” instead of “ailments”.

You can use “between” or “in” instead of “from“ for sentence with 1990 – 2017.

Materials and Methods

Ethics committee approval information should be included in this section in accordance with journal rules.

For the section 2.2.1., according to which scientific resources did you specify the process in the translation process. Give reference.

Results

You can explain the abbreviations required in the tables under the table.

Also, in tables with p-values, I recommend specifying significant values with a *.

Discussion

You said inclusion and exclusion criteria, but there is no information about these in the materials and methods section.

You did not provide the ICC value for the original language of the CRQ-SAS. What is the reason?

I suggest discussing why you prefer SGRQ over other questionnaires or assessment methods for validity.

In items 3, 4 and 5, you found low values in both factor analysis and test-retest relaibility. You should discuss the reasons for this and the possible effects during use of the questionnaire.

It may not be appropriate to start the last paragraph with "furthermore". You have started a new paragraph.

Last Paragraph of Discussion

“...can be used by the health professional to provide accurate prognostic outcomes.” In order to make this inference, you would have to evaluate responsiveness. As you stated in the following sentences, you did not look at responsiveness. So you can't comment on usage prognostically.

For completeness, it would be more accurate to use the abbreviation CRD instead of chronic respiratory disease.

As I suggested in the title revision, you only included COPD from many CRDs. Your analysis reveals the psychometric properties of the Urdu version of this questionnaire for COPD. Therefore, it is better to correct the study title as translation to Urdu language and validity and reliability in COPD. A similar usage was preferred in the Malay version of the questionnaire.

You may prefer the validity and reliability analyzes of the Urdu version for other diseases in future studies.

Conclusion

Such use can be made: “CRQ-SAS was translated to Urdu language with success. Furthermore this version is a valid and reliable tool for COPD patients who speaking Urdu.”

Reviewer #2: The manuscript on the Development of Urdu Version of Chronic Respiratory Disease Questionnaire Self-Administered Standardized (CRQ-SAS); Validity and Reliability Analysis showcases a well conducted study.

Here are a few of my remarks.

Generally, the manuscript is well structured, but there are a few grammatical (punctuations) and typographical errors. The authors needs to look at the entire paper again. E.g. COVID is spelled as COVD in the discussion and the conclusion sections is missing a number of words

There is a need to make the abstract methods and results section in synchrony... (eg. if reliability is coming before validity maintain the same flow in all the above listed sections accordingly).

No information on where the healthy participants were recruited from and if they were matched with the COPD patients

No justification for the number of participants recruited.

It wasn't stated where the 5 COPD patients used for cognitive debrief were sourced from.

Lastly, the discussion section is a bit too simplistic and shallow. Kindly dwell more on choice of the methods, clinical implication and limitations

Reviewer #3: Comments to the Author

Thanks for the opportunity to review “Development of Urdu Version of Chronic Respiratory Disease Questionnaire Self- Administered Standardized (CRQ-SAS); Validity and Reliability Analysis”. The authors described the translation process of CRQ-SAS and examined its psychometric properties. This is a well-written study. However, a few more details need to be included in the manuscript. The following are recommendations for the authors of the manuscript.

Abstract: Some abbreviations used for the first time should be written in long form (CRD, COPD).

Introduction:

1. I suggest the authors highlight why this tool is needed. How it differs from currently used assessment tools. These issues should be elaborated.

Material and Methods:

1. Is the sample size calculation done? It should be detailed.

2. In cognitive debriefing part COPD should be used as an abbreviation. The long form has been used before in the text.

3. In Proofreading and finalization of the translation part, It should be indicated which experts participated in the committee.

4. Method is clear, but I suggest the authors clarify the CRQ-SAS administration process. For example, Who interviewed the patients? Did they receive training? What are their backgrounds?

Result:

1. In Table 1, whether there is a difference between the groups in terms of demographic characteristics should be indicated with appropriate statistical analysis.

Discussion:

1.For convergent validity, other studies used assessments such as 6-minute walk distance (6MWD), short form-36 (SF-36), graded exercise test, and pulmonary function test, but you used only SGRQ. You should explain why you used only SGRQ in the discussion section.

2. Limitations of the study should be stated.

6. PLOS authors have the option to publish the peer review history of their article (what does this mean?). If published, this will include your full peer review and any attached files.

Reviewer #1: No

Reviewer #2: **Yes: **Jibril Mohammed

Reviewer #3: No

---

## [Author Response · Author response to Decision Letter 0]

15 Sep 2023

Comments from Reviewer Authors Response to Comments 

Journal requirements

1. Please ensure that your manuscript meets PLOS ONE's style requirements, including those for file naming. The manuscript is updated as per PLOS One Guidelines which includes the change in the heading sizes and Figures and Tables Captions.

2. We note that you have indicated that data from this study are available upon request. PLOS only allows data to be available upon request if there are legal or ethical restrictions on sharing data publicly.

b) If there are no restrictions, please upload the minimal anonymized data set necessary to replicate your study findings as either Supporting Information files or to a stable, public repository and provide us with the relevant URLs, DOIs, or accession numbers. I agree to upload the minimal anonymized data set necessary to replicate your study findings as Supporting Information files.

The Cover Letter remains the same as previously uploaded.

3. Your ethics statement should only appear in the Methods section of your manuscript. If your ethics statement is written in any section besides the Methods, please move it to the Methods section and delete it from any other section. Please ensure that your ethics statement is included in your manuscript, as the ethics statement entered into the online submission form will not be published alongside your manuscript. The ethics statement is now written in the methods sections and is also removed form Supporting Information section.

4. Please include a separate caption for each figure in your manuscript. Caption for each figures are mentioned in the in separate file (List of Figures)

5. Please upload a copy of Figure 3, to which you refer in your text on page 10. If the figure is no longer to be included as part of the submission, please remove all reference to it within the text. The caption of figure 5 was mistakenly written. There are three figures (Fig 1, Fig 2 & Fig 3) used in the manuscript which are uploaded in separate file (List of Figures)

6. Please review your reference list to ensure that it is complete and correct. If you have cited papers that have been retracted, please include the rationale for doing so in the manuscript text, or remove these references and replace them with relevant current references. Any changes to the reference list should be mentioned in the rebuttal letter that accompanies your revised manuscript. If you need to cite a retracted article, indicate the article’s retracted status in the References list and also include a citation and full reference for the retraction notice. The references list is complete and UpToDate. No references for the retracted papers have been cited in the manuscript. 

Reviewer #1

1. Thanks for the effort you put into your work. I consider that the study in question will be appropriate for publication after a few minor revisions. I recommend that you reconsider on a language basis, especially for some word usages. With this; The suggestions that I can make for you valuable researchers are listed below.

It can be given in the title as translation and its validity and reliability in COPD patients. For example; Translation of Chronic Respiratory Disease Questionnaire Self-Administered Standardized (CRQ-SAS) to Urdu Language and Validity and Reliability in COPD. I really appreciate the reviewer’s suggestion in changing the title of manuscript by mentioning COPD in it. But at this stage this will require reapproval form Ethics Committee of Riphah International University which is a tedious task now.

2. Write the long versions of the abbreviations for the first use. Such as COPD. In addition, if you have chosen an abbreviation, you should prefer the abbreviation rather than the clear version throughout the text, like CRD in discussion part of article. I have thoroughly checked the manuscript for the commonly used abbreviations like CRDs, COPD, SEM, SDC and B&A Plot. The first abbreviation is written with the full form with later on, the abbreviation are used alone.

3. Prefer to use CRQ-SAS U everywhere. In some places it remained CRQ SAS U. Indicate at first use that U is the Urdu version.

 The first full form of CRQ-SAS U is now stated in Line 98. From there on only abbreviation is used.

4. Minimal Detectable Change may be a more accurate use instead of Smallest Detectable Change.

I suggest that you review the overall text in terms of language and spelling. You have used the 2nd – 3rd meanings of some words. Smallest Detectable Change is replaced with Minimal Detectable Change. The overall text is also reviewed for language and spelling.

5. I suggest you replace some of the words you use with more common uses. For instance; “involve, contain” instead of “encompass”; “disorders” instead of “ailments”. You can use “between” or “in” instead of “from“ for sentence with 1990 – 2017. Both encompass and ailments words are replaced by their better alternatives as suggested by the reviewer.

6. Ethics committee approval information should be included in this section in accordance with journal rules. The ethics statement is now written in the methods sections and is also removed form Supporting Information section.

7. For the section 2.2.1., according to which scientific resources did you specify the process in the translation process. Give reference. The appropriate reference of the process of is cited now. (Line 98)

8. You can explain the abbreviations required in the tables under the table. Abbreviations are explained under table 1. The rest of the abbreviation used in the tables are already explained in the text of the manuscript

9. Also, in tables with p-values, I recommend specifying significant values with a * In table 5, the last column of P value is removed now and significant value are marked with *

10. You said inclusion and exclusion criteria, but there is no information about these in the materials and methods section. Inclusion and Exclusion Criteria is mentioned in the Material and Methods Section (Line 83 – 88)

11. You did not provide the ICC value for the original language of the CRQ-SAS. What is the reason? The ICC=0.83-0.95 short term Reliability of the original version of CRQ SAS

12. I suggest discussing why you prefer SGRQ over other questionnaires or assessment methods for validity. SGRQ is generic self-reported measure like CRQ SAS that can be easily used to determine the impact of CRDs on health-related quality of life. Unlike other subjective measures like COPD Assessment Test, Clinical COPD Questionnaire, this questionnaire is free to use and requires no permission to use. SGRQ also has similar subdomains like CRQ SAS and hence was also preferred over SF 36 which is a nonspecific QOL measure. Since the data was collected form the outpatient department with very small time margin of patient interaction, the use of objective measures like 6-minute walk distance (6MWD), graded exercise test, and pulmonary function test was not feasible. 

This is now mentioned in Discussion Section.

13. In items 3, 4 and 5, you found low values in both factor analysis and test-retest reliability. You should discuss the reasons for this and the possible effects during use of the questionnaire. CRQ SAS contains questions that inquires information related to the impact of disease on activities. Among these questions, item 3 (walking), item 4 (performing chores such as housework, shopping, groceries) & item 5 (participating in social activities) are directly linked to participation. As the socioeconomic status and daily roles of the participants was not considered, this might be the possible explanation for low ICC and item to total correlation of these three items only.

14. It may not be appropriate to start the last paragraph with "furthermore". You have started a new paragraph. The starting of the paragraph is now changed

15. “...can be used by the health professional to provide accurate prognostic outcomes.” In order to make this inference, you would have to evaluate responsiveness. As you stated in the following sentences, you did not look at responsiveness. So, you can't comment on usage prognostically. The sentence has been changed now 

16. For completeness, it would be more accurate to use the abbreviation CRD instead of chronic respiratory disease.

 I have thoroughly checked the manuscript for the commonly used abbreviations like CRDs, COPD, SEM, SDC and B&A Plot. The first abbreviation is written with the full form with later on, the abbreviation are used alone.

17. As I suggested in the title revision, you only included COPD from many CRDs. Your analysis reveals the psychometric properties of the Urdu version of this questionnaire for COPD. Therefore, it is better to correct the study title as translation to Urdu language and validity and reliability in COPD. A similar usage was preferred in the Malay version of the questionnaire. I really appreciate the reviewer’s suggestion in changing the title of manuscript by mentioning COPD in it. But at this stage this will require reapproval form Ethics Committee of Riphah International University which is a tedious task now.

18. You may prefer the validity and reliability analyzes of the Urdu version for other diseases in future studies. Its mentioned in the last paragraph of the discussion 

Reviewer #2

1. Generally, the manuscript is well structured, but there are a few grammatical (punctuations) and typographical errors. The authors need to look at the entire paper again. E.g. COVID is spelled as COVD in the discussion and the conclusion sections is missing a number of words The said mistakes in the spellings are now corrected. The manuscript is also proofread thoroughly for any other spelling mistakes, grammatical (punctuations) and typographical errors.

2. There is a need to make the abstract methods and results section in synchrony... (e.g. if reliability is coming before validity maintain the same flow in all the above listed sections accordingly). Abstract, Methods and Results Section are now synchronous. In all the sections the Reliability is mentioned first followed by Validity.

3. No information on where the healthy participants were recruited from and if they were matched with the COPD patients The information is now mentioned in Material and Methods Section (Line 83 – 88).

4. No justification for the number of participants recruited. The sample size justification is elaborated now in manuscript;

Sample size estimation was based on ICC value and width of confidence interval (95% CI) form previous study using the formula n = 16p(1−p)/w2, where p was lowest expected ICC and w was the maximum width of 95%CI. The minimum sample size estimated was 87, but we recruited a larger sample of 124 subjects (62 COPD & 62 healthy) to account for expected dropout at retest interval.

5. It wasn't stated where the 5 COPD patients used for cognitive debrief were sourced from. They were sourced from the same clinical settings from where the actual data was collected. I have stated this in manuscript now. 

6. Lastly, the discussion section is a bit too simplistic and shallow. Kindly dwell more on choice of the methods, clinical implication and limitations The discussion section is revised now in the light of the comments of reviewers.

Reviewer #3

1. Abstract: Some abbreviations used for the first time should be written in long form (CRD, COPD).

 I have thoroughly checked the manuscript for the commonly used abbreviations like CRDs, COPD, SEM, SDC and B&A Plot. The first abbreviation is written with the full form with later on, the abbreviations are used alone.

1. Introduction:

I suggest the authors highlight why this tool is needed. How it differs from currently used assessment tools. These issues should be elaborated. 

2. Material and Methods:

Is the sample size calculation done? It should be detailed. The sample size justification is elaborated now in manuscript;

Sample size estimation was based on ICC value and width of confidence interval (95% CI) form previous study using the formula n = 16p(1−p)/w2, where p was lowest expected ICC and w was the maximum width of 95%CI. The minimum sample size estimated was 87, but we recruited a larger sample of 124 subjects (62 COPD & 62 healthy) to account for expected dropout at retest interval.

3. In cognitive debriefing part COPD should be used as an abbreviation. The long form has been used before in the text.

 I have thoroughly checked the manuscript for the commonly used abbreviations like CRDs, COPD, SEM, SDC and B&A Plot. The first abbreviation is written with the full form with later on, the abbreviations are used alone.

4. In Proofreading and finalization of the translation part, It should be indicated which experts participated in the committee. The experts that participated in proof reading are mentioned now.

5. Method is clear, but I suggest the authors clarify the CRQ-SAS administration process. For example, Who interviewed the patients? Did they receive training? What are their backgrounds?

 The administration process is now mentioned under the Psychometric testing of the CRQ-SAS U

6. Result:

In Table 1, whether there is a difference between the groups in terms of demographic characteristics should be indicated with appropriate statistical analysis.

 Table I is modified as per the comments of the Reviewer.

7. Discussion:

Limitations of the study should be stated.

 Limitations are now mentioned at the end of Discussion Section.

8. For convergent validity, other studies used assessments such as 6-minute walk distance (6MWD), short form-36 (SF-36), graded exercise test, and pulmonary function test, but you used only SGRQ. You should explain why you used only SGRQ in the discussion section.

 SGRQ is generic self-reported measure like CRQ SAS that can be easily used to determine the impact of CRDs on health-related quality of life. Unlike other subjective measures like COPD Assessment Test, Clinical COPD Questionnaire, this questionnaire is free to use and requires no permission to use. SGRQ also has similar subdomains like CRQ SAS and hence was also preferred over SF 36 which is a nonspecific QOL measure. Since the data was collected form the outpatient department with very small time margin of patient interaction, the use of objective measures like 6-minute walk distance (6MWD), graded exercise test, and pulmonary function test was not feasible. 

This is now mentioned in Discussion Section.

---

## [Decision Letter · Decision Letter 1]

2 Oct 2023

PONE-D-23-14764R1Development of Urdu Version of Chronic Respiratory Disease Questionnaire Self-Administered Standardized (CRQ-SAS); Validity and Reliability AnalysisPLOS ONE

Dear Dr. Hassan,

Thank you for submitting your manuscript to PLOS ONE. After careful consideration, we feel that it has merit but does not fully meet PLOS ONE’s publication criteria as it currently stands. Therefore, we invite you to submit a revised version of the manuscript that addresses the points raised during the review process.

We look forward to receiving your revised manuscript.

Kind regards,

Gurkan Gunaydin

Academic Editor

PLOS ONE

Journal Requirements:

Additional Editor Comments :

Dear Authors

The 2nd revision process of your article has been completed. While 2 of the reviewers stated that the article was suitable for publication, the other reviewer requested a minor correction. Please make the necessary correction and upload the article to the system again.

Regards

Reviewers' comments:

Reviewer's Responses to Questions

**Comments to the Author**

1. If the authors have adequately addressed your comments raised in a previous round of review and you feel that this manuscript is now acceptable for publication, you may indicate that here to bypass the “Comments to the Author” section, enter your conflict of interest statement in the “Confidential to Editor” section, and submit your "Accept" recommendation.

Reviewer #1: (No Response)

Reviewer #2: All comments have been addressed

Reviewer #3: (No Response)

2. Is the manuscript technically sound, and do the data support the conclusions?

Reviewer #1: Yes

Reviewer #2: Yes

Reviewer #3: Yes

3. Has the statistical analysis been performed appropriately and rigorously? 

Reviewer #1: Yes

Reviewer #2: Yes

Reviewer #3: Yes

4. Have the authors made all data underlying the findings in their manuscript fully available?

Reviewer #1: Yes

Reviewer #2: Yes

Reviewer #3: Yes

5. Is the manuscript presented in an intelligible fashion and written in standard English?

Reviewer #1: Yes

Reviewer #2: Yes

Reviewer #3: Yes

6. Review Comments to the Author

Reviewer #1: Thank you for your revisions. There are a few things I would like to add regarding revisions. I recommend you check them out.

Why did you change the author name and order after revision in the study? You mentioned that it is difficult to deal with the ethics committee for the title change. You should explain the reason for the change in author names without applying to the ethics committee. I think it would be appropriate to send your ethics committee approval document to editor. However, I think the title revision is absolutely necessary. Even in the conclusion part, you specified the result from the study “…we recommend health care professionals to use this tool to measure the disease-related quality of life in native COPD patients.”. It is sufficient for ethical compliance that the study method and participants are the same as in your ethics committee application. The one word change you make in the title does not create an ethical problem. I leave the final decision to the editor.

My other review recommendations are listed below. If you make the revisions, I think your article is suitable for publication. The main point that needs to be clarified is that the author name and names order change. It must be proven that this has been done with ethics committee approval.

Capitalize the first letters in the survey names on line 70.

You should remove the phrase "progression" from the sentence in lines 316-320. You did not evaluate responsiveness.

You attributed the low values in the factor analysis and reliability analysis results for items 3, 4 and 5 to socioeconomic level and lifestyle. “Walking (item 3), Performing chores, such as housework, shopping or grocery shopping (item 4), Participating in social activities, such as meeting with family, friends (item 5)”. You should explain the relationship of these items to the inference you mentioned. It is not clearly understood how walking is associated with socioeconomic level and lifestyle. Also, discuss this statement and add it to the discussion.

Reviewer #2: The revised version of the paper is now better written and contains all the information i earlier asked. I wish to thank the authors for a wonderful revision

Reviewer #3: The requested revisions have been made. It is appropriate for me to publish it in this form. Kind regards.

7. PLOS authors have the option to publish the peer review history of their article (what does this mean?). If published, this will include your full peer review and any attached files.

Reviewer #1: No

Reviewer #2: **Yes: **Jibril Mohammed

Reviewer #3: No

---

## [Author Response · Author response to Decision Letter 1]

13 Oct 2023

Why did you change the author name and order after revision in the study? You mentioned that it is difficult to deal with the ethics committee for the title change. You should explain the reason for the change in author names without applying to the ethics committee. I think it would be appropriate to send your ethics committee approval document to editor. However, I think the title revision is necessary. Even in the conclusion part, you specified the result from the study “…we recommend health care professionals to use this tool to measure the disease-related quality of life in native COPD patients.”. It is sufficient for ethical compliance that the study method and participants are the same as in your ethics committee application. The one word change you make in the title does not create an ethical problem. I leave the final decision to the editor.

My other review recommendations are listed below. If you make the revisions, I think your article is suitable for publication. The main point that needs to be clarified is that the author name and names order change. It must be proven that this has been done with ethics committee approval. The Ethics Approval Document is already shared with the Editor of PLOS One in the very first attempt to submit the article.

This article is derived from Thesis work of a student named Momina Kashif (1st Author in the article), with me (Danish Hassan, 2nd Author) being her supervisor. The Ethical Review Committee Letter is issued by the name of Momina Kashif and the committee does not mention any other name on it. Hence the Ethical Committee does not regulate the names and their sequence that are meant to be written on manuscript produced form the thesis work. The Ethical Committee Letter is attached at the end for your instance also. 

In reply to your second question about why the authors name was changed at this time is that here are 7 people that had intellectually contributed to finalization of this manuscript. I entertained only the first 5 in this manuscript based on the volume of their contribution. The author Hira Humayun has disagreed to financially contribute to this manuscript and asked for the removal of her name from it and hence I added a new name in replacement of her.

I have also changed the Title of Manuscript as per your recommendation.

Capitalize the first letters in the survey names on line 70. The correction is done in line 70

You should remove the phrase "progression" from the sentence in lines 316-320. You did not evaluate responsiveness. The word is now removed

You attributed the low values in the factor analysis and reliability analysis results for items 3, 4 and 5 to socioeconomic level and lifestyle. “Walking (item 3), Performing chores, such as housework, shopping or grocery shopping (item 4), Participating in social activities, such as meeting with family, friends (item 5)”. You should explain the relationship of these items to the inference you mentioned. It is not clearly understood how walking is associated with socioeconomic level and lifestyle. Also, discuss this statement and add it to the discussion. The patients that participated in this study were heterogenous in terms of the daily activities that they performed, occupation and life roles. Social activities vary across the socioeconomic status and walking is also linked to the nature of occupation might be the possible explanation for low item to total correlation of these three items only. 

Socioeconomic status is the main factor that decides what sort of roles a person will have in this part of the world. For instance, managers of the company will have different walking patterns as compared to lower staff who has walk as a part of their job duty. The same is applicable to the other two factors (performing chores, such as housework, shopping, or grocery shopping & participating in social activities.

Seprate legends for Each Figures are now updated in Manuscript

---

## [Decision Letter · Decision Letter 2]

24 Oct 2023

Development of Urdu Version of Chronic Respiratory Disease Questionnaire Self-Administered Standardized (CRQ-SAS); Validity and Reliability Analysis in COPD patients

PONE-D-23-14764R2

Dear Dr. Hassan,

We’re pleased to inform you that your manuscript has been judged scientifically suitable for publication and will be formally accepted for publication once it meets all outstanding technical requirements.

Kind regards,

Gurkan Gunaydin

Academic Editor

PLOS ONE

Additional Editor Comments (optional):

Dear Authors

The revision phase of your article has been completed. Your article has been deemed suitable for publication.

Reviewers' comments:

Reviewer's Responses to Questions

**Comments to the Author**

1. If the authors have adequately addressed your comments raised in a previous round of review and you feel that this manuscript is now acceptable for publication, you may indicate that here to bypass the “Comments to the Author” section, enter your conflict of interest statement in the “Confidential to Editor” section, and submit your "Accept" recommendation.

Reviewer #1: All comments have been addressed

2. Is the manuscript technically sound, and do the data support the conclusions?

Reviewer #1: Yes

3. Has the statistical analysis been performed appropriately and rigorously? 

Reviewer #1: Yes

4. Have the authors made all data underlying the findings in their manuscript fully available?

Reviewer #1: Yes

5. Is the manuscript presented in an intelligible fashion and written in standard English?

Reviewer #1: Yes

6. Review Comments to the Author

Reviewer #1: Dear Authors,

Thank you for your information about the name and its order. Please do not take what I wrote for ethical approval personally. Since it is a sensitive issue, I wanted to focus on it. Since you paid attention to the revision suggestions and made the necessary clarifications and corrections, I give an acceptance opinion to your manuscript.

Best Regards.

7. PLOS authors have the option to publish the peer review history of their article (what does this mean?). If published, this will include your full peer review and any attached files.

Reviewer #1: No

---

## [Editor Report · Acceptance letter]

18 Dec 2023

PONE-D-23-14764R2 

PLOS ONE

Dear Dr. Hassan, 

I'm pleased to inform you that your manuscript has been deemed suitable for publication in PLOS ONE. Congratulations! Your manuscript is now being handed over to our production team.

Kind regards, 

on behalf of

Assoc. Prof. Gurkan Gunaydin 

Academic Editor

PLOS ONE